# Discovery of Carcinogenic Liver Fluke Metacercariae in Second Intermediate Hosts and Surveillance on Fish-Borne Trematode Metacercariae Infections in Mekong Region of Myanmar

**DOI:** 10.3390/ijerph17114108

**Published:** 2020-06-09

**Authors:** Ei Ei Phyo Myint, Amornpun Sereemaspun, Joacim Rocklöv, Choosak Nithikathkul

**Affiliations:** 1Tropical and Parasitic Diseases Research Unit, Graduate Studies Division, Faculty of Medicine, Mahasarakham University, Mahasarakham 44000, Thailand; dreepm85@gmail.com; 2Department of Anatomy, Faculty of Medicine, Chulalongkorn University, Bangkok 10330, Thailand; anatomy_tan@hotmail.com; 3Department of Public Health and Clinical Medicine, Section of Sustainable Health, Umeå University, SE–901 87 Umeå, Sweden; joacim.rocklov@umu.se

**Keywords:** opisthorchiidae, *opisthorchis viverrini*, minute intestinal flukes (MIF), Heterophyidae

## Abstract

Countries of lower Mekong regions are highly alarmed by the spread of fish-borne trematode infections, i.e., small liver flukes and minute intestinal flukes especially in Thailand, Lao People’s Democratic Republic (Lao PDR), Vietnam, Cambodia and Myanmar. Moreover; the incidence of cholangiocarcinoma has also been increasing in the endemic area of liver fluke infections. Only a few reports have been published concerning the fish-borne trematodes infections in the central region of Myanmar. However; there is still a lack of information regarding the status of trematodes infections in second intermediate hosts in the Mekong region of Myanmar. Therefore, we conducted surveillance on the distribution of trematode metacercariae in small cyprinoid fishes collected from the Mekong region of Myanmar. A total of 689 fishes (12 different species of cyprinoid fishes) have been collected and examined by pepsin digestion methods. We discovered four species of fish-borne trematode metacercariae infections, i.e., carcinogenic liver fluke, *Opisthorchis viverrini*; minute intestinal flukes, *Haplorchis taichui; Haplorchis pumilio* and *Haplorchoides* sp. in Tachileik, the Mekong Region of Myanmar. The outcome of this study could be a useful index for the fish-borne zoonotic trematode epidemiology in the Mekong area. Besides, the results of our study contribute to filling the gap of information necessary for the control and prevention of fish-borne trematode zoonotic infections in the Mekong region.

## 1. Introduction

The important public health problem of the spread of fish-borne trematodes (FBT) has highly alarmed the countries of the lower Mekong basin especially Thailand, Lao People’s Democratic Republic (Lao PDR), Cambodia, Vietnam and Myanmar [1,2,3,4,5]. Fish-borne zoonotic trematodes (FZT) especially small liver flukes (Opisthorchiidae) and minute intestinal flukes (Heterophyidae) are highly prevalent in those regions [6]. These two flukes have similar life cycles involving two intermediate hosts [7,8]. The first intermediate hosts are snails, and the second intermediate host are small freshwater fishes [9,10]. Humans are infected through ingestion of undercooked or raw freshwater fishes, which are contaminated with the infective stage of the parasite, i.e., metacercariae [11]. The traditional habit of eating raw or undercooked fish is a known risk factor for human trematode infections [7]. Opisthorchiasis caused by carcinogenic liver fluke, *Opisthorchis viverrini,* is an important public health problem in lower Mekong region countries [12,13,14,15,16,17,18,19]. Heterophyiasis is an infection of the small bowel, which is caused by minute intestinal flukes (Heterophyidae) [20,21,22,23]. The incidence of cholangiocarcinoma has also been increasing in the endemic area of liver fluke infections [24,25]. In Myanmar, there is one report in 2019 about a retrospective study of the cholangiocarcinoma cases admitted to the Hepatobiliary and Pancreatic surgery department, Yangon Specialty Hospital, Myanmar, which shows that the prevalence has notably been increasing since 2016, and the highest prevalence was in 2018 [26]. Myanmar people also have the traditional habit of eating raw small cyprinoid fishes pickled with rice, locally called (Nyar lay Chin) (Figure 1). Moreover, infection can also occur via poor personal hygiene and the contamination of food, hands and food preparation utensils that are contaminated with metacercariae [27,28,29,30]. For the increasing data of cholangiocarcinoma cases from Myanmar, the associated risk factors such as carcinogenic liver fluke infection and epidemiological data require more surveillance for the control and prevention. In the central region of Myanmar, seven species of minute intestinal flukes (members of the Heterophyidae) have been reported in Yangon, Myanmar [5]. However, there is still a lack of information about the fish-borne trematode infection status from the Mekong region of Myanmar. Tachileik is a town situated in the Mekong basin of Myanmar (Figure 2). Geographically, the city is also situated in the Golden Triangle that is the area where the borders of Thailand, Laos and Myanmar meet at the confluence of the Ruak and Mekong rivers. Therefore, we conducted a survey on trematode matercerceriae infections in second intermediate hosts from Tachileik, Mekong region of Myanmar, and we believe that our data will provide required epidemiological information to fill the knowledge gap and provide valuable data to solve the public health problem of control and prevention of fish-borne trematode zoonotic infections in countries of the Mekong basin.

## 2. Material and Methods

### 2.1. Collection of Cyprinoid Fishes and Identification of Fish Species

A total of 689 fishes (12 different species of cyprinoid fishes) were purchased from local markets in Tachileik, lower Mekong region of Myanmar (Table 1). Tachileik is a town situated in the Mekong basin of Myanmar (Figure 2), which is also situated in the Golden Triangle that is the area where the borders of Thailand, Laos and Myanmar meet at the confluence of the Ruak and Mekong rivers. The cyprinoid fishes were collected from January 2018 to December 2019. All collected fishes were transferred on ice to the laboratory of the Department of Medicine, Mahasarakhum University. The length and width of fishes were individually measured, and the species of all fishes were identified with the aid of the Fishbase website (http://www.fishbase.org/search.php) (Table 1). 

### 2.2. Pepsin Digestion Method

Examination of zoonotic trematode metacercariae in the second intermediate hosts was done by pepsin-HCl artificial digestion techniques. All the collected fishes were ground one by one in a mortar with pestle, and then, the ground samples were transferred into a beaker and mixed with artificial gastric juice. The artificial gastric juice is made by a mixture of 10 gm pepsin A, 10 ml of concentrated HCl and 0.85% NaCl or normal saline 1000 mL [31]. The grounded samples were mixed well and were placed in a 37 °C incubator for 2 h with occasional stirring and removed the larger particles by the filtration of digested materials. Then 0.85% saline was added to the digested sample, and we let it stand for a while and discarded the supernatant very carefully and kept the sediment. The procedures were repeated 8 or 9 times until the supernatant became clear and a small quantity of the sediment was transferred into a Petri dish containing 6–7 ml physiological saline after which metacercariae were observed and identified using a stereomicroscope and light microscope. The detected metacercariae were isolated and were put into a small dish. Finally, the number of metacercariae of each fluke species were counted for further analysis.

### 2.3. Identification of Metacercariae 

For the identification of metacercariae, first of all, similar-shaped metacercariae were collected separately based on the general morphological features in a small Petri dish. Secondly, they were moved with a spoid onto a glass slide and were covered with a coverslip; then, detailed morphology was observed under a light microscope. Finally, the metacercariae were identified based on characteristic morphological features. As the characteristic features, the shape of cysts, size of suckers, shape and contents of the excretory bladder were identified [32,33].

### 2.4. Analysis of Findings

After collected metacercariae were categorized according to the size and morphological characteristics, their percentage prevalence was calculated as follows:Prevalence (%) = Number of infected fish × 100/Total number of fish examined(1)

Intensity is the number of metacercariae per fish infected [5].

## 3. Results

### 3.1. Infection Status and Prevalence of Fish-Borne Trematode Infection in Freshwater Fishes from Tachileik, Mekong Region of Myanmar

A total of 689 fishes (12 different species of cyprinoid fishes) were collected from local markets in Tachileik, Mekong region of Myanmar. In the total of 689 fishes, 391 fishes were infected with the fish-borne trematode infections, and the overall prevalence was 46.29%. Among the 12 different species, 8 species of cyprinoid fishes, i.e., *B. gonionotus*, *P. falcifer*, *M. marginatus*, *C. repasson*, *S. Rubripinnis*, *l. siamensis, H. siamensis*, *R. argyrotaenia,* were contaminated with the infective stage of trematode parasites, metacercariae (Table 2). The metacercariae of *O. viverrini* were found in *C. repasson* (2.64%, *n* = 151). The metacercariae of *H. taichui* were found infecting 4 species among the total 12 species of fishes. The highest infection prevalence among the four species of fish was found in *R. argyrotaenia* (45%, *n* = 20), and then, in decreasing order, in *M. marginatus* (38%, *n* = 100), *C. repasson* (35%, *n* = 151) and *L. siamensis* (20%, *n* = 60). The metacercariae of *H. pumilio* were infected in 5 out of 12 species of fishes. The highest infection prevalence among the four species of fishes was found in *B. gonionotus* (65%, *n* = 20) and then with decreasing order in *Systomus rubripinnis* (60.71%, *n* = 28), *P. falcifer* (53.33%, *n* = 30), *M. marginatus* (14%, *n* = 100) and *C. repasson* (7.94%, *n* = 151). The metacercariae *Haplorchoides* sp. were present in 5 out of 12 species of fishes. The highest prevalence was detected in *C. repasson* (82.78%, *n* = 151), and the lowest prevalence was occurred in *H. siamensis* (56%, *n* = 50) and the others were found in *L. siamensis* (63.33%, *n* = 60), *M. marginatus* (68%, *n* = 100) and *R. argyrotaenia* (70%, *n* = 20), respectively.

### 3.2. Fish-Borne Trematode Metacercarial Intensity in Cyprinid Fishes from Tachileik, Mekong Region of Myanmar

A total of 11 metacercariae of *O. viverrini* were detected from *C. repasson* with a mean intensity of 2.75 per fish infected (Table 3). The metacercariae of *H. taichui* were collected in 94 (28.39%) out of 331 fishes (4 species), i.e., *L. siamensis, M. marginatus, R. argyrotaenia, C. repasson,* with a mean intensity of 3.75 per fish infected (Table 4). The metacercariae of *H. pumilio* were detected in 72 (21.88%) out of 329 fishes (5 species), i.e., *B. gonionotus, P. falcifer, S. rubripinnis, M. marginatus, C. repasson,* with a mean intensity of 4.11 per fish infected (Table 5). The metacercariae of *Haplorchoides* sp. were detected in 273 (71.65%) out of 381 fishes (5 species), i.e., *L. siamensis*, *H. siamensis*, *M. marginatus, R. argyrotaenia*, *C. repasson*, with a mean intensity of 3.57 per fish infected (Table 6).

### 3.3. Co-infections of Trematode Metacercariae in Fresh Water Cyprinoid Fishes Collected from Tachileik, Mekong Region of Myanmar

Simultaneous infections of one fish with two or more trematode species were recorded in some fish species, i.e., *L. siamensis*, *M. marginatus, R. argyrotaenia*, *C. repasson* (Table 2). Four species of fish-borne trematode metacercariae, i.e., *O. viverrini*, *H. taichui*, *H. pumilio*, *Haplorchoides* sp. simultaneously infected *Cyclocheilichthys repasson* (Table 2). The fish species *M. marginatus* was infected with *H. taichui*, *H. pumilio*, *Haplorchoides* sp. (Table 2). Co-infection of trematode metacercariae *H. taichui, Haplorchoides* sp. was observed in *L. siamensis* and *R. argyrotaenia* (Table 2).

### 3.4. Morphology of Detected Metacercariae of Fish-Borne Trematode Infections in Freshwater Fishes from Tachileik, Mekong Region of Myanmar

Metacercariae of *O. viverrini* were elliptical, had nearly equal sized oral sucker and ventral sucker, brownish pigment granules scattered within the body and an O-shaped excretory bladder occupying the greater part of the posterior body (Figure 3a). *H. taichui* metacercariae were elliptical and had a baseball glove-shaped ventrogenital sac with rodlets and an O-shaped excretory bladder occupying large portion of posterior body (Figure 3b). *H. pumilio* metacercariae were elliptical and had deer horn-like minute spines arranged in 1–2 rows around the ventrogenital complex and an O-shaped excretory bladder occupying large portion of posterior body (Figure 3c). *Haplorchoides* sp. are nearly spherical, with a double layered cystic wall and have lance-shaped bodies, with a scale like spine on the body surface. Acetabulum with spines present and excretory bladder is saccular (Figure 3d).

## 4. Discussion

Small freshwater cyprinid fishes are the second intermediate host of trematode infections. Two major agents of fish-borne infections are liver flukes (Opisthorchiidae) and intestinal flukes (Heterophyidae). There are many reports of zoonotic trematode infected fishes that are found in lower Mekong basin countries especially in Thailand [34,35,36,37], Laos [38,39], Cambodia [40,41,42,43], Vietnam [44,45,46,47] and the central region of Myanmar [48,49,50]. Nevertheless, the contaminated fish species are different in each region. In our study, a total of twelve species of cyprinoid fishes, i.e., *B. gonionotus*, *P. falcifer*, *S. rubripinnis*, *L. siamensis*, *H. siamensis*, *M. marginatus*, *R. argyrotaenia*, *S. Orphoides*, *P. Brevis*, *T. trichopterus*, *C. repasson* and *A. testudineus*, were collected and examined by pepsin digestion methods. We discovered four species of fish-borne trematode metacercariae infections, i.e., carcinogenic liver fluke, *O. viverrini*; minute intestinal flukes, *H. taichui, H. pumilio* and *Haplorchoides* sp. in Tachileik, Mekong Region of Myanmar. 

Chronic infection caused by the Southeast Asian liver fluke (*O. viverrini*) is a critical risk factor for the development of the bile duct cancer cholangiocarcinoma (CCA), which is a major public health concern in Mekong region countries [12,13,14,15,16]. *O. viverrini* infection in humans occurs via the consumption of raw or uncooked fish, which contains metacercariae. A better understanding of the epidemiology of these fish-borne parasites is important for the prevention of CCA in the community. In our study, we discovered the presence of *O. viverrini* metacercariae in small freshwater cyprinoid fish, *C. repasson*, from Tachileik, Mekong region of Myanmar. In addition, this finding is the first report of carcinogenic liver fluke, *O. viverrini* metacercariae infection from Mekong region of Myanmar. The spread of *O. viverrini* infections relies on the presence of intermediate hosts, i.e., first intermediate host (snails), the second intermediate hosts (freshwater fishes) and dietary preferences of local populations for uncooked or undercooked fish. Traditional dishes prepared from freshwater cyprinoid fishes are the source of fish-borne trematode infections [36]. Therefore, local people should be educated about the danger of ingestion of undercooked fish dishes, in order to increase awareness for the prevention of fish-borne trematode transmission. 

In Myanmar, seven species of minute intestinal flukes, i.e., *H. taichui, H. pumilio, H. yokogawai, Centrocestus* sp., *Stellantchasmus falcatus*, *Pygidiopsis cambodiensis*, and *Procerovum* sp., were reported in fishes from local market of Yangon, Myanmar in 2017 by J. Chai et al. [5]. They found that *H. taichui* metacercariae were detected in 5 species of fishes, *T. thynnoides, P. aurotaeniatus, E. altus, Mystacoleucus* sp. and *Labeo* sp. [5]. In our present study, 4 out of 12 species of freshwater fishes were found positive for *H. taichui* metacercariae, i.e., *M. marginatus*, *C. repasson*, *L. siamensis* and *R. argyrotaenia.* Therefore, a total of 9 species of freshwater fishes from Myanmar are confirmed to be second intermediate hosts of *H. taichui* metacercariae. Currently, *H. taichui* metacercariae have been identified as occurring in 48 species of freshwater cyprinoid fishes in southeast Asia countries [51,52,53,54]. 

In addition, we found that *H. pumilio* metacercariae infected 5 out of 12 species, i.e., *B. gonionotus, P. falcifer*, *M. marginatus*, *C. repasson* and *S. rubripinnis.* In the study from J. Chai et al. 2017, *H. pumilio* metacercariae were detected in 9 species of freshwater fishes, *T. thynnoides*, *P. aurotaeniatus*, *E. altus, C. striata*, *A. testudineus*, *Rhynogobius* sp., *T. pectoralis, Mystacoleucus* sp. and *Labeo* sp. [5]. Hence, a total of 14 species of freshwater fishes from Myanmar are the second intermediate hosts of *H. pumilio* metacercariae. Currently, a total of 39 freshwater cyprinoid fish species have been recorded as the second intermediate host of *H. pumilio* in southeast Asia countries [55,56,57,58]. 

The metacercariae *Haplorchoides* sp. were infected in 5 species among the total 12 species, i.e., *L. siamensis*, *H. siamensis, M. marginatus*, *R. argyrotaenia* and *C. repasson.* The finding of *Haplorchoides* sp. from our study is the first reported from Myanmar. In 2018, K. Apiwong et al. reported finding *Haplorchoides* sp. in *B. schwanenfeldii* and *C. repasson* from Chiang Mai province, Thailand [59]. Our study confirms that the life cycle of *Haplorchoides* sp. happens around Tachileik, the Mekong Region of Myanmar. 

## 5. Conclusions

In this study, four species of trematode metacercariae (carcinogenic liver flukes, *O. viverrini,* and minute intestinal flukes, members of the Heterophyidae, *H. taichui*, *H. pumilio* and *Haplorchoides* sp.) were detected in freshwater fishes from a local market of Tachileik, the Mekong region of Myanmar. This study demonstrates the existence of the life cycles of four species of fish-borne trematode infections around Tachileik, the Mekong region of Myanmar. The study of the distribution and infection status of the fish-borne trematode metacercariae could contribute to solving the important public health problem posed by these trematodes and may provide valuable information for prevention and control programs of human liver fluke and intestinal fluke infections for the community. Furthermore, the outcome of this study could be a useful index in the trematode epidemiology in the Mekong area. In addition, the results of our study contribute important information for collaborative prevention of carcinogenic liver fluke infection among the lower Mekong region countries. Further research studies should be considered for public health interventions using health-education and sanitation-improvement approaches in the control program. The findings of our present study would be important and supportive not only from a parasitological but also from a public-health point of view.

## Figures and Tables

**Figure 1 ijerph-17-04108-f001:**
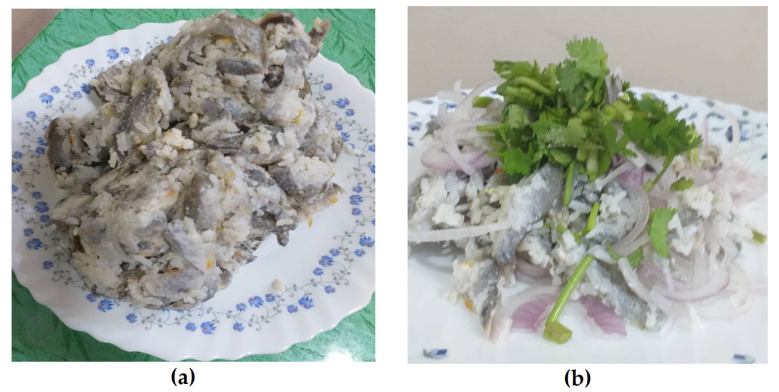
Photos of local Myanmar food (**a**) Raw small cyprinoid fishes pickled with rice (Nyar lay Chin) (**b**) Nyar lay Chin prepared as an uncooked salad mixed with onion (Nyar lay Chin salad).

**Figure 2 ijerph-17-04108-f002:**
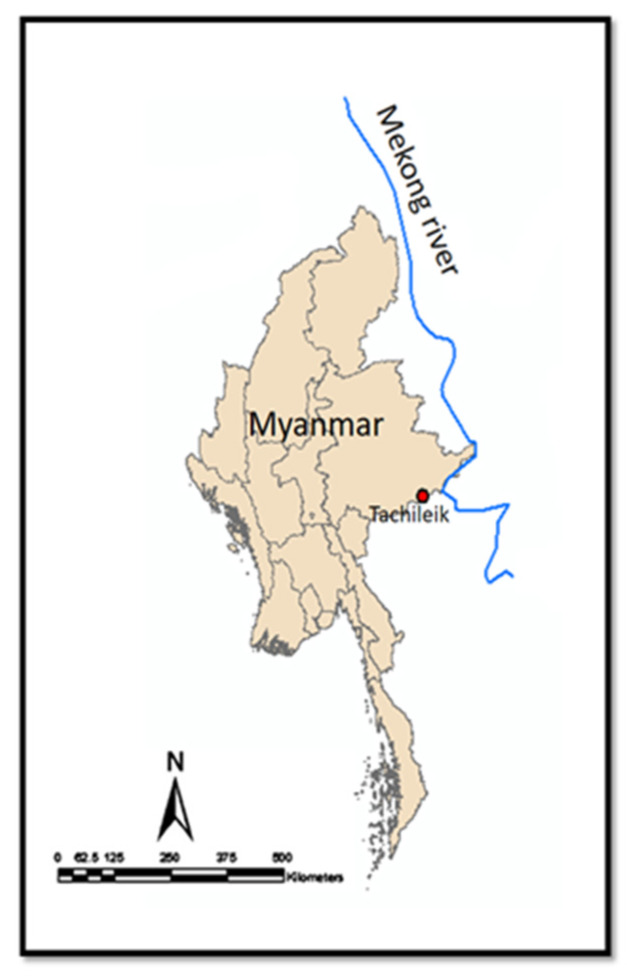
Geographic mapping of freshwater cyprinid fishes collected area from Tachileik, Mekong region of Myanmar (20°27′ N 99°53′ E) by ArcGIS 10.5.

**Figure 3 ijerph-17-04108-f003:**
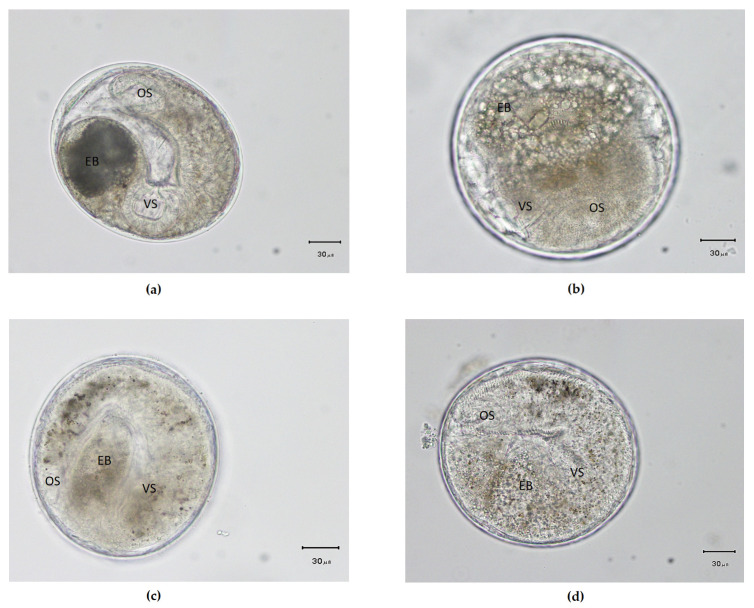
Photos of fish-borne trematode metacercariae detected in freshwater cyprinoid fishes from Tachileik, Mekong region of Myanmar. (**a**) *O. viverrini* (**b**) *H. taichui* (**c**) *H. pumilio* (**d**) *Haplorchoides* sp. (EB, excretory bladder; OS, oral sucker; VS, ventral sucker).

**Table 1 ijerph-17-04108-t001:** Freshwater cyprinoids fishes purchased from the local market in Tachileik, Mekong region of Myanmar.

Number	Fish Species	Total Number	Average Length	Average Width
1	*Barbonymus gonionotus*	20	9.5 cm ± 0.2236	3 cm ± 0.3244
2	*Puntioplites falcifer*	30	9.5 cm ± 0.2170	3 cm ± 0.2110
3	*Systomus rubripinnis*	28	9 cm ± 0.3502	3 cm ± 0.2377
4	*Labiobarbus siamensis*	60	10 cm ± 0.219	3.3 cm ± 0.3188
5	*Henicorhynchus siamensis*	50	12.5 cm ± 0.2175	3.5 cm ± 0.3023
6	*Mystacoleucus marginatus*	100	9.5 cm ± 0.2564	3.3 cm ± 0.3128
7	*Rasbora argyrotaenia*	20	10.5 cm ± 0.2236	3.5 cm ± 0.3244
8	*Systomus orphoides*	10	12.5 cm ± 0.9368	3.5 cm ± 0.3496
9	*Puntius brevis*	100	10.5 cm ± 0.2422	3.5 cm ± 0.2983
10	*Trichogaster trichopterus*	20	11.5 cm ± 0.2236	3.5 cm ± 0.2511
11	*Cyclocheilichthys repasson*	151	9.5 cm ± 0.2230	3.5 cm ± 0.2804
12	*Anabas testudineus*	100	9 cm ± 0.2240	3 cm ± 0.2876
	total	689		

**Table 2 ijerph-17-04108-t002:** Infection status and prevalence of fish-borne trematode infections in freshwater fish from Tachileik, Mekong region of Myanmar.

No	Fish Species	No. of Fish Examined	No. (%) of Fish Infected with FBT	No. (%) of Fish Infected with Ov	No. (%) of Fish Infected with Ht	No. (%) of Fish Infected with Hp	No. (%) of Fish Infected with Hap
1	*Barbonymus gonionotus*	20	13 (65)	_	_	13 (65)	_
2	*Puntioplites falcifer*	30	16 (53.33)	_	_	16 (53.33)	_
3	*Systomus rubripinnis*	28	17 (60.71)	_	_	17 (60.71)	_
4	*Labiobarbus siamensis*	60	38 (63.33)	_	12 (20)	_	38 (63.33)
5	*Henicorhynchus siamensis*	50	28 (56)	_	_	_	28 (56)
6	*Mystacoleucus marginatus*	100	68 (68)	_	38 (38)	14 (14)	68 (68)
7	*Rasbora argyrotaenia*	20	14 (70)	_	9 (45)	_	14 (70)
8	*Systomus orphoides*	10	0 (0)	_	_	_	_
9	*Puntius brevis*	100	0 (0)	_	_	_	_
10	*Trichogaster trichopterus*	20	0 (0)	_	_	_	_
11	*Cyclocheilichthys repasson*	151	125 (82.78)	4 (2.64)	35 (23.17)	12 (7.94)	125 (82.78)
12	*Anabas testudineus*	100	0 (0)	_	_	_	_
	Total	689	319 (46.29)				

FBT, Fish-borne trematode; Ov, *Opisthorchis viverrini*; Ht, *Haplorchis taichui*; Hp, *Haplorchis pumilio*; Hap, *Haplorchoides* sp.

**Table 3 ijerph-17-04108-t003:** Intensity of *O. viverrini* metacercariae detected in freshwater fishes from Tachileik, Mekong region of Myanmar.

No	Fish Species	No. of Fish Examined	No. (%) of Fish Infected Ov	Total Metacercariae Detected	Range (Min–Max)	Intensity
1	*Cyclocheilichthys repasson*	151	4 (2.64)	11	(2–3)	2.75

Ov, *O. viverrini*.

**Table 4 ijerph-17-04108-t004:** Intensity of *H. taichui* metacercariae detected in freshwater fishes from Tachileik, Mekong region of Myanmar.

No	Fish Species	No. of Fish Examined	No. (%) of Fish Infected Ht	Total Metacercariae Detected	Range (Min–Max)	Intensity
1	*Labiobarbus siamensis*	60	12 (20)	102	(3–12)	8.50
2	*Mystacoleucus marginatus*	100	38 (38)	120	(2–10)	3.15
3	*Rasbora argyrotaenia*	20	9 (45)	10	(1–2)	1.11
4	*Cyclocheilichthys repasson*	151	35 (23.17)	121	(1–9)	3.45
	Total	331	94 (28.39)	353	(1–12)	3.75

Ht, *H. taichui*.

**Table 5 ijerph-17-04108-t005:** Intensity of *H. pumilio* metacercariae detected in freshwater fishes from Tachileik, Mekong region of Myanmar.

No	Fish Species	No. of Fish Examined	No. (%) of Fish Infected Hp	Total Metacercariae Detected	Range (Min–Max)	Intensity
1	*Barbonymus gonionotus*	20	13 (65)	24	(1–4)	1.84
2	*Puntioplites falcifer*	30	16 (53.33)	16	(1)	1
3	*Systomus rubripinnis*	28	17 (60.71)	59	(1–7)	3.47
4	*Mystacoleucus marginatus*	100	14 (14)	67	(1–6)	4.78
5	*Cyclocheilichthys repasson*	151	12 (7.94)	130	(1–21)	10.83
	Total	329	72 (21.88)	296	(1–21)	4.11

Hp, *Haplorchis pumilio*.

**Table 6 ijerph-17-04108-t006:** Intensity of *Haplorchoides* sp. metacercariae detected in freshwater fishes from Tachileik, Mekong region of Myanmar.

No	Fish Species	No. of Fish Examined	No. (%) of Fish Infected Hap	Total Metacercariae Detected	Range (Min–Max)	Intensity
1	*Labiobarbus siamensis*	60	38 (63.33)	102	(1–7)	2.68
2	*Henicorhynchus siamensis*	50	28 (56)	43	(1–3)	1.53
3	*Mystacoleucus marginatus*	100	68 (68)	120	(1–5)	1.76
4	*Rasbora argyrotaenia*	20	14 (70)	16	(1–2)	1.14
5	*Cyclocheilichthys repasson*	151	125 (82.78)	695	(1–21)	5.56
	Total	381	273 (71.65)	976	(1–21)	3.57

Hap, *Haplorchoides* sp.

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
