# Peer review of "Discovery of Carcinogenic Liver Fluke Metacercariae in Second Intermediate Hosts and Surveillance on Fish-Borne Trematode Metacercariae Infections in Mekong Region of Myanmar"

_ijerph, 2020, doi:10.3390/ijerph17114108_

Round 1

Reviewer 1 Report

Fish borne diseases are of major importance for public health. The problem of human health risks due to fish ingestion is an important issue as foodborne parasitic infections are frequent worldwide due to wild and aquaculture fish. The present manuscript describes infections of fish by trematode metacercariae with zoonotic potential which is important.

Introduction

In the text, writing in the first person plural should be avoided (eg. line 44 … we also have the traditional…).

More information on the zoonotic potential of these parasites should be added in order the importance of the study to be highlighted.

Information about the study area (lines 57-60) should be added at the materials and methods section.

The purpose of the study (lines 60-64) should be rephrased.

Materials and methods

In the Table 1 standard error should be added at the mean values.

Figure 3 is not in accordance with the text.

In the identification of metacercariae section (lines 99-104), passive voice should be used.

More information about the calculation of intensity should be added. The figure 4 describes mean intensity and not the number of parasites per total number of fish.

Results

Standard error should be added at the mean values.

Figure 3 in the text should be replaced by figure 4.

More information about the organs where the parasites were detected should be added.

Discussion

Discussion needs further improvement and the text is poor. The applied aspect of this study should be added in the discussion. More information should be highlighted regarding the zoonotic potential of these parasites and the impact on public health, supported by references. Also, guidelines regarding food safety issues should be mentioned.

Reviewer 2 Report

Please see review in attachment.

Round 2

Reviewer 1 Report

The manuscript is improved according to reviewers comments. 

Author Response

I have changed according to reviewers comments in the manuscript.

Reviewer 2 Report

The authors have made the required changes and the manuscript is significantly improved. In my opinion, the manuscript is now in a condition to be published. I recommend, however, that the authors make the following small corrections to the text:

line 54 – case → cases

line 55 – need to be more surveillance → require more surveillance

lines 55–56 – for the control and prevention program. → for their control and prevention.

line 63 – second intermediate host → second intermediate hosts

line 67 – countries of Mekong basin → countries of the Mekong basin

lines 74–75 situated in Mekong basin → situated in the Mekong basin

line 116 - cyprinoids fishes → cyprinoid fishes

line 122 - The metacercariae of H. taichui were infected in 4 species → The metacercariae of H. taichui were found infecting 4 species

line 125 - The metacercariae H. pumilio were infected in 5 out of 12 species of fishes → The metacercariae of H. pumilio were present in 5 out of 12 species of fishes

lines 128–129 - The metacercariae Haplorchoides sp. were infected in 5 out of 12 species of fishes → The metacercariae Haplorchoides sp. were present in 5 out of 12 species of fishes.

lines 142–143 – an average density of 2.75 per fish infected → a mean intensity of 2.75

lines 144–145 - with their average intensity was 3.75 per fish infected → with a mean intensity of 3.75

line 147 – with their average density of 4.11 per fish infected → with a mean intensity of 4.11

line 149 – with their average density of 3.57 per fish infected → with a mean intensity of 3.57

Tables 3,4,5,6 → Perhaps you might want to combine these into a single larger table? It’s not essential, but it might look better in the final published version.

line 204 - consumption of a raw or uncooked fish → consumption of raw or uncooked fish

line 209 - is relied on → relies on

lines 212–214 – Therefore, local people should be educated for the danger of ingestion of undercooked fish dishes and increased awareness for the prevention of fish borne trematode transmission. → Therefore, local people should be educated about the danger of ingestion of undercooked fish dishes, in order to increase awareness for the prevention of fish borne trematode transmission.

Several places: fish borne → fish-borne

Author Response

I have changed according to reviewers comments.

We would like to present as an individual species' intensity data in table 3,4,5,6.